# Protocol for Transcranial Direct Current Stimulation for Obsessive-Compulsive Disorder

**DOI:** 10.3390/brainsci10121008

**Published:** 2020-12-18

**Authors:** Peta E. Green, Andrea Loftus, Rebecca A. Anderson

**Affiliations:** School of Psychology in Faculty of Health Sciences, Curtin University, Kent Street, Bentley, WA 6102, Australia; andrea.loftus@curtin.edu.au (A.L.); rebecca.anderson@curtin.edu.au (R.A.A.)

**Keywords:** OCD, transcranial direct current stimulation (tDCS), non-invasive brain stimulation, neuromodulation, obsessive, compulsive, protocol, randomised controlled trial

## Abstract

Obsessive-compulsive disorder (OCD) is a debilitating disorder with an approximate lifetime prevalence of 1–3%. Despite advances in leading treatment modalities, including pharmacotherapy and psychotherapy, some cases remain treatment resistant. Non-invasive brain stimulation has been explored in this treatment-resistant population with some promising findings; however, a lack of methodological rigor has reduced the quality of the findings. The current paper presents the protocol for conducting research into the efficacy of transcranial direct current stimulation (tDCS) in the treatment of OCD. A double-blind randomised controlled trial (RCT) will be conducted involving active tDCS vs. sham tDCS on 40 general OCD patients. The intervention consists of 2 mA anodal stimulation over the pre-supplementary motor area (pre-SMA) with the cathode positioned over the orbitofrontal cortex (OFC). Participants will receive 10 sessions of 20 min of either sham- or active-tDCS over 4 weeks. Outcomes will be categorical and dimensional measures of OCD, as well as related secondary clinical measures (depression, anxiety, quality of life), and neurocognitive functions (inhibitory control, cognitive flexibility).

## 1. Introduction

The frontostriatal model suggests that the symptoms of obsessive-compulsive disorder (OCD) are associated with dysfunction in the feedback loop, which leads to hyperactivity in the orbitofrontal cortical (OFC) pathways. As a result, individuals pay more attention to threatening stimuli. Neurological studies indicate that OCD symptoms are associated with increased neuronal activity in the OFC [1,2,3] and decreased activation in the pre-supplementary motor area (pre-SMA) [4] responsible for inhibitory control. Changes in patterns of activation (both hyperactivity and hypoactivity) in the frontal and motor cortices have been reported in a range of neuropsychiatric disorders, including ADHD and schizophrenia [5]. It has been suggested that an imbalance in cortical activation may underlie OCD [6,7,8]. With this in mind, several studies have attempted to address this imbalance in neuronal activation using non-invasive brain stimulation.

Transcranial direct current stimulation (tDCS) is a non-invasive stimulation technique used to modulate cortical activity and, as a consequence, change behaviour [9]. tDCS delivers low intensity electrical currents to modulate neuronal activity [10]. tDCS works by applying a positive (anodal) or negative (cathodal) current via electrodes to a targeted brain area. The current modifies cortical excitability by facilitating the depolarisation or hyperpolarisation of neurons respectively. Research in healthy individuals has found that anodal (excitatory) tDCS over frontal areas improves cognitive functioning [11,12]. Lawrence [13] reported that anodal tDCS over frontal areas led to improved cognition in those with Parkinson’s disease.

Although there is limited research in brain stimulation interventions for OCD, preliminary studies involving tDCS for patients defined as resistant to both CBT and pharmacotherapy have been encouraging. Hazari [7] conducted an open-label study using anodal tDCS over pre-SMA to increase pre-SMA activation, and with the cathode over OFC to decrease OFC activation. The patient, who was on escitalopram and clonazepam, received 2 mA of current for 20 min, twice a day for 10 days (20 sessions). This tDCS montage was based on the theory that OCD is due to deficient pre-SMA response inhibition on frontostriatal function [4,14]. The patient demonstrated an 80% reduction in OCD symptom severity, which was maintained for 7 months post-intervention with minor fluctuations. Whilst the patient did relapse at the 7-month time point, their symptoms improved following a further 8 sessions of tDCS. On both occasions that tDCS was applied, the patient’s OCD symptoms had remitted within 5–10 days. Narayanaswamy [8] used the same tDCS protocol combined with therapeutic SSRIs in two patients, who received 2 mA of current for 20 min, twice a day for 5 days (10 sessions). Both patients demonstrated significantly reduced OCD symptom severity (52% and 46.7% reduction) that was maintained at 1 and 2-month follow-up assessments. However, despite a few studies reporting positive effects of tDCS on OCD, the efficacy of non-invasive brain stimulation remains ambiguous [6], and there is a lack of consensus regarding stimulation protocols in regard to frequency, dose, intensity, and electrode montage/positioning. Further, there are several limitations associated with the published studies. 

Many of the tDCS and OCD studies published are single-patient case studies, and only two have included more than 10 participants [15,16]; thus, the results are not generalisable. No study to date has included a sham-control group. Few studies used an OCD symptom measure as the primary outcome, and those that did not indicate whether a change in the OCD symptom measure was clinically significant. Other aspects of functioning such as quality of life, inhibitory control, and cognitive flexibility have not been explored. Brunelin [6] conducted a systematic review of studies examining tDCS for those with OCD and concluded that although a number of studies demonstrated improvements in OCD, there was a lack of methodological rigor that reduced the quality of the findings.

## 2. Materials and Methods

The proposed clinical trial (ACTRN12620000990921) will be a double-blind randomised sham-controlled trial of tDCS for OCD to examine the therapeutic potential of tDCS for OCD symptoms. To the best of our knowledge, there has not been a double-blind sham-control randomised controlled trial (RCT) of tDCS for OCD, limiting conclusions about the effectiveness of this treatment approach.

### 2.1. Hypotheses

We hypothesise that active tDCS over the pre-SMA and OFC will be associated with a clinically and statistically significant decrease in OCD symptoms and beliefs, a significant decrease in comorbid depression and anxiety symptoms (if present), and a significant increase in quality of life. We also propose there will be improved inhibitory control and cognitive flexibility.

### 2.2. Participants

#### 2.2.1. Sampling

Recruitment of participants will occur through several channels. Primary recruitment will occur via the Curtin University Psychology Clinic, which houses a specialist OCD service, whereby promotional material will be distributed to suitable clients. A media release will be made through Curtin University, and private clinics specialising in OCD will be contacted/informed about the project and sent copies of advertising materials. Individuals who wish to take part in the study will follow instructions on the promotional material to contact the researcher. Interested participants will then be sent an information pack and consent form. Potential participants will be telephone-screened for tDCS suitability and will undergo a risk assessment in relation to the exclusion criteria prior to commencing the study. Participants who meet the inclusion criteria will then be contacted by the researcher to confirm their willingness to participate, inform them of start dates, offer them the opportunity to ask questions, and arrange the pre-treatment (baseline) assessment of outcomes. Following the post-intervention assessments (after session 10), a time will be scheduled for the 3-month and 6-month follow-up outcome assessments.

#### 2.2.2. Inclusion Criteria

Participants aged over 18-years with obsessive and/or compulsive behaviours who meet the criteria for the clinical diagnosis of OCD 300.3 (F42.2) in accordance with the Diagnostic and Statistical Manual of Mental Disorders (DSM-5; [17], using the Mini-International Neuropsychiatric Interview—Version 7.0.0 (MINI) [18].A minimum Yale-Brown Obsessive-Compulsive Scale (Y-BOCS) total score of 16, representing a minimum of moderate symptom severity.Participants taking medications (SRI/SSRIs) to manage OCD symptoms will be included as long as the dose has been stable for at least 12 weeks prior to participation [19] and they do not plan to change the dose during the study.

#### 2.2.3. Exclusion Criteria

Participants with a recent history of brain surgery, neurological conditions associated with brain abnormalities (e.g., traumatic brain injury; recent stroke, tumour), implanted cranial devices, hearing aids (unless they can be removed), or active skin lesions on the scalp.History of migraine, epilepsy, seizures, unstable medical and/or psychiatric conditions; history of psychosis or bipolar disorder; high suicide/self-harm risk.Current or past (within the last 1 month) use of benzodiazepines, anticonvulsants, lithium carbonate, psychostimulants, dextromethorphan and pseudoephedrine; recreational drug use.Currently undergoing Exposure and Response Prevention (ERP) therapy for OCD; any neuromodulation therapy (e.g., ECT, transcranial magnetic stimulation, tDCS) within the last 3 months.

#### 2.2.4. Randomisation

Participants will be randomly allocated to one of two treatment groups through block randomisation [20]: (1) active tDCS, or (2) sham tDCS. A computer-generated randomisation list will be used to allocate participants to groups at a ratio of 1:1 in blocks of four.

#### 2.2.5. Blinding

The investigators (researcher and supervisors) and the participants will be blinded to the group allocation. Each of the 40 participants will be allocated a personal identification (ID) number and be randomly assigned to either the active or the sham condition. The administrator (a third party) will create 40 protocols, each with the participant’s ID number, and load them onto the device. The procedure will be password-protected, and the researcher (operator) will not have access to the password. Once the double-blind mode is activated on the device, all non-essential information is hidden on the monitor, keeping the operator unaware of which protocol refers to active stimulation and which protocol is sham. When a participant arrives for their session, the operator will select the protocol number that corresponds to the participant’s ID number on the tDCS device and will not know which condition they have been allocated to.

## 3. Study Design

Stimulation will take place in the Curtin Psychology Clinic, where tDCS will be administered by the researcher. All participants will complete ten 20-min sessions of either active or sham tDCS stimulation over 4 weeks (all outcome-measures to be conducted pre- and post-intervention, and at 3- and 6-month follow-ups) (Table 1).

Participants will be seated in a comfortable armchair whilst receiving tDCS, which will be delivered for 20 min using a battery-driven (Necbox) multi-channel direct current stimulator, namely Starstim 20^TM^ (Neuroelectrics, Barcelona, Spain). Stimulation will be administered over the scalp via two 25 cm^2^ Sponstim electrodes (Neuroelectrics, Barcelona, Spain). The electrodes consist of a sponge cover, a carbon rubber core, and a nickel-plated brass metallic pin. The external surface of the sponges will be soaked in 5 mL of 0.9% saline solution to minimise the risk of skin irritation, and inserted into Fz (pre-SMA) and Fp1 (OFC) of an adult-sized neoprene cap (S, M, or L), which is pre-labelled according to the 10–20 EEG system of electrode positioning. The participant’s head will be measured to locate Cz (mid-line central part of the head), and then the cap will be placed on the participant’s head. Once the cap is in the correct position (with Cz lined up), it will be secured in place with a chin strap, and the medical sockets will be connected to the electrodes. An impedance check will be conducted to ensure optimal conductivity. If the impedance level is high, more saline solution will be added onto the surface of the sponges by inserting a curved syringe through a hole in the cap near the electrode.

Each session will involve 20 min of either active or sham stimulation. The tDCS montage (stimulation site, intensity, duration, and frequency) is informed by Kekic’s 2016 systematic review [21]. The participants in the active group will receive a constant current 2 mA stimulation via the anode placed over the pre-SMA and cathode placed over the left OFC. The anode will increase neural activation of the pre-SMA and the cathode will decrease neural activation of the left OFC (Figure 1). For the sham stimulation, the electrode montage will be identical to the active tDCS group; however, the participants will only receive tDCS for 30 s at the start, ramp up (0–2 mA) and end, ramp down (2–0 mA) of the session. This allows the participants to experience some sensation of tDCS. All participants (active and sham) will be informed that they may or may not perceive any sensation during the treatment, a procedure that has been demonstrated to effectively blind participants and the researcher to the stimulation condition they are in [22].

### 3.1. Outcome Measures

Diagnostic screening, neuropsychological, and cognitive assessment measures for this study will be administered at baseline, post-intervention, 3-month-, and 6-month follow-up. The Y-BOCS, Depression Anxiety Stress Scales (DASS-21), and Obsessive Beliefs Questionnaire (OBQ-44) will also be administered at the end of the 1st, 2nd, and 3rd week of the intervention period to identify if, and when, changes may occur. OCD symptom change, as measured by the Y-BOCS, is the primary outcome being assessed in this study.

Diagnostic screening: The MINI-7.0.0 [20] will be used for clinical diagnosis of OCD.

OCD symptom severity: The Y-BOCS [23] measures symptom type and severity over the last 7-day period and consists of two 10-item subscales, namely obsessions and compulsions. 

Negative affect: The Depression Anxiety Stress Scales (DASS-21) [24] is a self-report measure to identify and measure negative affect (if present).

Quality of life: The Quality of Life Enjoyment and Satisfaction Questionnaire-Short Form (Q-LES-Q-SF) [25], also a self-report measure, has 16 items evaluating overall enjoyment and satisfaction with physical health, mood, work, household and leisure activities, social and family relationships, daily functioning, sexual life, economic status, overall well-being, and medications.

OCD beliefs: The Obsessive Beliefs Questionnaire (OBQ-44) measures OCD beliefs. The OBQ-44 includes (1) perfectionism and intolerance of uncertainty, (2) importance and control of thoughts, (3) responsibility, and (4) overestimation of threat, which are positively associated with obsessive-compulsive symptoms and worry [26].

Inhibitory function: The Stop Signal Task is a choice go/no-go reaction-time task to measure inhibitory control. In this computerised task, participants are required to discriminate between left and right arrows by pressing the appropriate response key as fast as possible (go), but inhibit their motor response if a beep is played after the presentation of the arrow (no-go) [27]. The task is designed so that the frequency of the ‘go’ cues are greater than the ‘no-go’ cues, resulting in the ‘go’ response becoming prepotent and thus control being required to inhibit/withhold the response [28].

Cognitive flexibility: The set-shifting task will be used to measure cognitive flexibility. Set-shifting measures the ability to shift attention and respond to a particular aspect of a stimulus depending on a reinforced contingency. The rules or contingencies of the task change and alternate rapidly, requiring the participant to pay attention and respond with the pertinent rule in mind, switching from the old to the new rule. Individuals with OCD demonstrate repetitive and perseverate behaviour, and impairments in set-shifting ability have been reported to be a key neurocognitive feature of OCD [28,29].

### 3.2. Potential Side Effects

The procedure is considered very low risk, and no significant adverse events have been reported with low-current procedures such as the ones proposed in this study. Potential side effects of low-current tDCS include localised scalp itching or tingling sensation at the site where the electrode was placed, and seldom-occurring headache or fatigue [30]. If any of these side effects occur, the participant will be monitored; if they do not dissipate within 1 hour (the typical duration of symptoms/s), the participant will be referred for assessment by a medical practitioner. Any adverse side effects will be reported to the appropriate ethics committee. Discontinuation and/or withdrawal from the study will be recorded in the study database.

### 3.3. Data: Sample Size, Management, Analysis

This study will provide evidence for the efficacy of the treatment approach, which may in turn lead to effectiveness trials for clients with more complex or particular OCD profiles. There are no suitable tDCS trials to guide a power analysis for this study; however, a G*Power calculation indicated that 30 participants (15 per group) are required to detect a moderate effect (α = 0.05; power = 0.80). We will aim to recruit 40 participants (20 per group) to allow for attrition.

A series of generalised linear mixed models (GLMMs), one for each of the seven outcome measures, will be used to determine whether active and sham tDCS differ at pre- versus post-intervention in the outcome measures. The GLMMs will be conducted using the GENLINMIXED procedure in SPSS (Version 26). GLMMs are used to control for outcome variables when the data are not normally distributed and include random and fixed effects [31]. This study has one nominal random effect (participant) and one nominal fixed effect (group: tDCS vs. sham), one ordinal fixed effect (time: pre-intervention, post-intervention, and 3- and 6-month follow-up), and the group x time interaction. GLMMs are robust against unequal groups [32], and unlike ANOVAs, GLMMs do not rely on participants providing data at each assessment point, reducing the effect of participant attrition on statistical power.

This study will conform to the guidelines under Section 2 of the Australian Code for Responsible Conduct of Research. All hard data (diagnostic screening, consent forms, psychometric measures) will be stored in a clinic file as part of the current clinic procedures. Data will be extracted and stored in a de-identified manner separate from the clinic file. Deidentified data will be shared if required by a journal. All electronic data will be password-protected, stored on the Curtin research drive, and backed up on an external hard drive, which will be kept in a locked drawer. Data will be kept for a minimum of 25 years, as per the Western Australian University Sector Disposal Authority (WAUSDA) guidelines for clinical trials, and only the researcher and supervisors will have access to data.

### 3.4. Ethics and Dissemination

This study will be conducted in accordance with the National Health and Medical Research Council [33] and the Code of Ethics [34]. Ethical approval has been granted by the Curtin University Human Research Ethics Committee (HRE2020-0266). All participants will provide written informed consent following a verbal and written explanation of the study protocol and the opportunity to ask any questions. Participants will be informed that participation is voluntary and that they have the right to withdraw at any time without question [33,34]). Results will be presented at conferences and reported in international peer-reviewed journals.

## 4. Discussion

This paper presents the protocol of a study designed to explore the efficacy and advance the knowledge of tDCS as a potential therapy for OCD. Current evidence-based treatments for OCD are pharmacotherapy and/or psychotherapy; however, considerable issues associated with tolerance and/or resistance to these treatments and subsequent relapse have led to a call for an alternative approach. Non-invasive brain stimulation has been explored in treatment-resistant cases, yielding some promising findings. However, the efficacy of tDCS as a treatment option remains ambiguous because of a lack of methodological rigour and clarity. We believe this will be the first double-blind randomised controlled trial to assess the efficacy of tDCS as a novel treatment intervention for OCD. This study will inform whether there is sufficient evidence of a treatment effect to progress to effectiveness trials and/or a larger, multicentre RCTs, which have higher costs and greater potential participant burden, since half of all participants receive a sham treatment that is likely ineffective.

## 5. Trial Status

This trial is actively recruiting and is expected to be completed (including follow-up outcome assessments) by July 2022.

## Figures and Tables

**Figure 1 brainsci-10-01008-f001:**
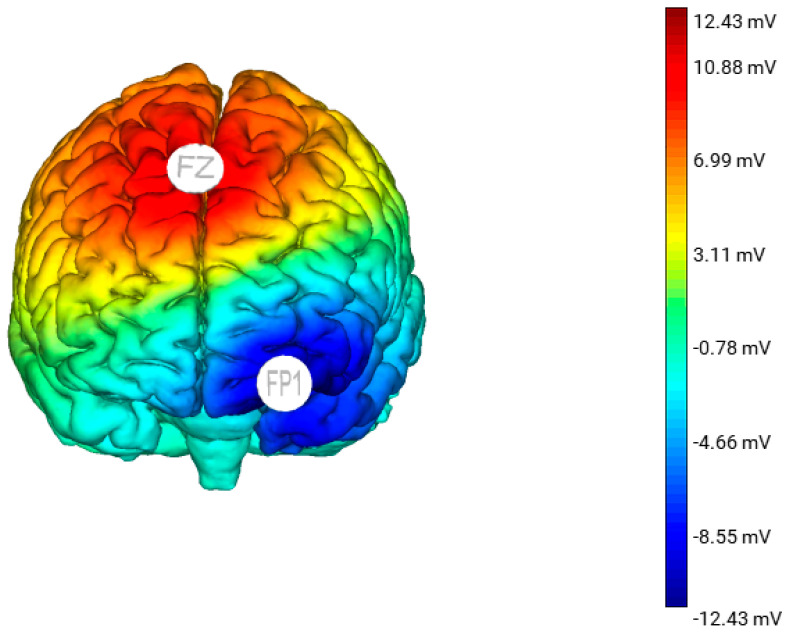
A 3D standardised model of the estimated electric field generated from anodal stimulation over Fz with the cathode placed over Fp1. This model was produced using the Neuroelectric Stim 20^TM^ Preview function.

**Table 1 brainsci-10-01008-t001:** Transcranial direct current stimulation (tDCS) protocol and assessment of outcome measures.

Outcome Measures	Week 1			Week 2			Week 3			Week 4			Follow-Up	
	Pre-Intervention Assessment	10 Sessions of tDCS (Active or Sham)										Post-Intervention Assessment	3-mth	6-mth
		S1	S2	S3	S4	S5	S6	S7	S8	S9	S10			
	T1		T2			T3			T4			T5	T6	T7
MINI	x											x	x	x
Y-BOCS	x		x			x			x			x	x	x
DASS 21	x		x			x			x			x	x	x
OBQ-44	x		x			x			x			x	x	x
Q-LES-Q-SF	x											x	x	x
Stop-Sig	x											x	x	x
Set-Shifting	x											x	x	x

Note. tDCS stimulation protocol for the active and sham groups involves 10 sessions conducted over 4 weeks. Participants attend the clinic three times per week with all outcome measures (x) assessed at 4 timepoints (T). Pre-intervention/baseline on day 1, post-intervention 2 days after the final tDCS session (S) S10, and follow-up measures at 3 and 6 months (mth). The Yale-Brown Obsessive Compulsive Scale (Y-BOCS), Depression Anxiety Stress Scales (DASS-21), and Obsessive Beliefs Questionnaire (OBQ-44) will also be administered at the end of week 1 (T2), week 2 (T3), and week 3 (T4) of the intervention period to monitor changes in symptom severity and/or sub-domains.

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
