# Peer review of "Protocol for Transcranial Direct Current Stimulation for Obsessive-Compulsive Disorder"

_brainsci, 2020, doi:10.3390/brainsci10121008_

Round 1

Reviewer 1 Report

Dear Authors,

Your paper is new and very interesting. Despite this, I reject your publication as I believe that a research article should have a results section that allows the scientific community to discuss and compare your work. It is a research protocol for which no efficacy data are provided. In addition, the protocol suffers from some threats to the validity of the randomised case study. It lacks a control group, the medication variable with NRS/RSRIs is not adequately controlled as six weeks are insufficient for some of these molecules to have the expected effect. The sample size is very small which threatens the internal validity of the study. There is no point in publishing a protocol in a scientific journal if it is not accompanied by the results of its application. The discussion section is underdeveloped; the protocol described is not compared with that of other research studies that are cited in the introduction. You must correct all these essential aspects in order for your research to be publishable. Publishing just to have a bigger resume does not make sense. The purpose of publications is to bring novel results to the fields of study.

Kind regards,

Author Response

Thank you for taking the time to review our manuscript and provide feedback. We have responded to and addressed each of the points you have raised. "Please see the attachment"

Reviewer 2 Report

“However, despite a few studies reporting positive effects of tDCS on OCD, the efficacy 57 of non-invasive brain stimulation remains ambiguous [15, 16]”  references concern Parkinson disease and not OCD

Inclusion criteria: I would add. OCD Patients suffering from at least 1 year

In my opinion, ten 20-minute sessions are not enough to get an improvement in OCD patients; how the authors justify this number of sessions whith ten 20-minute sessions which may minim ten 20-minute sessions procedure which may minimize the outcome at the end of the ten 20-minute sessions

“There are no suitable tDCS trials to guide a power analysis for this study, 202 however, a G*Power calculation indicated that 30 participants (15 per group) are required to detect a 203 moderate effect (α = .05; power = .80). We will aim to recruit 40 participants (20 per group) to allow 204 for attrition” . I don’t understand authors considering that the trial is neither a proof of concept study (open trial) nor a real randomized control trial with a sample size big enough to clearly respond to the issue. A multi site randomized trial involving more then 100 cases-control would be wiser in my opinion. Once again as we saw with rTMS,  it’s Diy. I understand that this type of trial is costly but this worth the pain.  

Author Response

Thank you for taking the time to review our manuscript and provide feedback and recommendations. We have responded to and addressed each of the points you have raised. "Please see the attachment".

Reviewer 3 Report

There is a true need for evidence-based, novel treatment options for OCD. tDCS is a relatively unexplored brain stimulation option that may prove to have real therapeutic benefit in this population. The protocol is generally well written and all-inclusive of the necessary information. Minor comments are below.

For ease of the reader, in figure 1 adding the specific assessments conducted at certain time points may be of benefit. 

I wonder if a mid-treatment (after treatment 5) batch of 'outcome' assessments should be done. Treatments could be efficacious quickly (and/or in different domains) and it would be of benefit to know how quickly symptom reduction is achieved. Maybe five treatments is just as efficacious as ten? Also, please collect 'duration of OCD illness.' Longer illness duration may help to explain a non-response.

OCD is not always the only psychiatric diagnosis in a presenting patient. Are psychotic and/or manic/bipolar patients with co-morbid OCD also eligible for enrollment? Alternatively, what about a past history of psychosis or mania? Furthermore, any drugs that attenuate brain/synaptic plasticity (e.g., anticonvulsants, benzodiazepines) may mitigate treatment effects. In Canada, we also have to keep a lookout for marijuana use, due to recent legalization, and this is also an exclusion criteria in several of our psychiatric clinical trials. 

Also, I would encourage the use of repeated-measures ANOVA over a GLMM. There are several ways to handle missing data that are appropriate to this experimental design. Furthermore, non-parametric statistical tests (e.g., Friedman Test) can be employed when data are not normally distributed. A repeated-measures ANOVA is robust (even in the face of assumption violations). In the end, a simplified statistical approach helps with interpretation and data transparency.  

I look forward to the findings from this exciting trial!

Author Response

Thank you for taking the time to review our manuscript, provide feedback and offer recommendations. We have responded to and addressed each of the points you have raised. "Please see the attachment."

Round 2

Reviewer 2 Report

Thank you for the added corrections . I apologize but I think that the primary criteria of assessement has not been defined by the autors ; this is crucial for the future paper presenting the results

Author Response

Thank you for alerting us to this.

We have added a sentence to the Outcome Measures section (Line 181) as follows to identify the primary outcome measure being assessed for this study: “OCD symptom change, as measured by the Y-BOCS, is the primary outcome being assessed in this study.”